# Prevalence and predictors of spontaneous preterm births in Nepal: findings from a prospective, population-based pregnancy cohort in rural Nepal–a secondary data analysis

Seema Subedi ,[1,2] Elizabeth A Hazel ,[1] Diwakar Mohan,[1] Scott Zeger,[3] Luke C Mullany,[1] James M Tielsch,[4] Subarna K Khatry,[2] Steven C LeClerq,[1,2] Robert E Black,[1] Joanne Katz[1]

For numbered affiliations see end of article.

**Correspondence to**
Seema Subedi;
ssubedi2@jhu.edu

## ABSTRACT

**Objective** Preterm birth can have short-term and long-term complications for a child. Socioeconomic factors and pregnancy-related morbidities may be important to predict and prevent preterm births in low-resource settings. The objective of our study was to find prevalence and predictors of spontaneous preterm birth in rural Nepal.

**Design** This is a secondary observational analysis of trial data (registration number NCT01177111).

**Setting** Rural Sarlahi district, Nepal.

**Participants** 40 119 pregnant women enrolled from 9 September 2010 to 16 January 2017.

**Outcome measures** The outcome variable is spontaneous preterm birth. Generalized Estimating Equations Poisson regression with robust variance was fitted to present effect estimates as risk ratios.

**Result** The prevalence of spontaneous preterm birth was 14.5% (0.5% non-spontaneous). Characteristics not varying in pregnancy associated with increased risk of preterm birth were maternal age less than 18 years (adjusted risk ratio=1.13, 95% CI: 1.02 to 1.26); being Muslim (1.53, 1.16 to 2.01); first pregnancy (1.15, 1.04 to 1.28); multiple births (4.91, 4.20 to 5.75) and male child (1.10, 1.02 to 1.17). Those associated with decreased risk were maternal education >5 years (0.81, 0.73 to 0.90); maternal height ≥150 cm (0.89, 0.81 to 0.98) and being from wealthier families (0.83, 0.74 to 0.93). Pregnancy-related morbidities associated with increased risk of preterm birth were vaginal bleeding (1.53, 1.08 to 2.18); swelling (1.37, 1.17 to 1.60); high systolic blood pressure (BP) (1.47, 1.08 to 2.01) and high diastolic BP (1.41, 1.17 to 1.70) in the third trimester. Those associated with decreased risk were respiratory problem in the third trimester (0.86, 0.79 to 0.94); having poor appetite, nausea and vomiting in the second trimester (0.86, 0.80 to 0.92) and third trimester (0.86, 0.79 to 0.94); and higher weight gain from second to third trimester (0.89, 0.87 to 0.90).

**Conclusion** The prevalence of preterm birth is high in rural Nepal. Interventions that increase maternal education may play a role. Monitoring morbidities during antenatal

## STRENGTHS AND LIMITATIONS OF THIS STUDY

⇒ This is a large population-based study that allows for analysis of rare and common risk factors for a relatively rare outcome (preterm).

⇒ Previous studies on preterm birth in Nepal were hospital based, enrolled women during delivery and have explored only the women's sociodemographic factors associated with preterm birth, whereas our study is population based, enrols women from earlier in pregnancy, follows them monthly, and has explored symptom and morbidity variables that change through pregnancy.

⇒ Gestational age at outcome has been measured using date of last menstrual period (LMP) as usually done in low/middle-income countries; however, as LMP was asked at enrolment that was generally early in pregnancy, there is less recall bias than LMP recalled at delivery or late in pregnancy.

⇒ Missing data for second trimester morbidities due to late enrolment of some women in pregnancy are a limitation, but comparison of sociodemographic characteristics suggests limited potential for biases due to this limitation.

care to intervene to reduce them through an effective health system may help reduce preterm birth.

## INTRODUCTION

Preterm birth (PTB) is defined as a birth occurring before 37 completed gestational weeks or fewer than 259 days from a woman's last menstrual period (LMP).[1] In 2010, the global prevalence of PTB estimated in 92 countries was 11.1% (95% CI: 9.1% to 13.4%), ranging from about 5% in some European countries to 18% in some African countries.[2] Sixty per cent of these PTBs occurred in sub-Saharan Africa and South Asia.[2] Complications of PTB were the

BMJ

leading causes of under-5 mortality and accounted for approximately 17.7% of all under-5 mortality and 36.1% of neonatal mortality, according to the 2019 global estimates.[3] Eighty-one per cent of the under-5 deaths from complications of PTB occurred in Asia and sub-Saharan African countries.[4]

PTBs can have short-term and long-term consequences. Short-term consequences comprise increased risks of neonatal respiratory conditions, sepsis, neurological conditions, feeding difficulties, and visual and hearing problems.[5–7] As the child grows, long-term consequences include more hospital admissions, poorer neurodevelopment outcomes, difficulties in learning, as well as behavioural and socioemotional problems.[8–10] At the family level, PTB can lead to significant economic and psychological difficulties, and at the national level, it leads to significant cost for the health system.[11 12]

In Nepal, under-5 mortality has dropped from 64 deaths to 39 deaths per 1000 live births (LBs) from 2001 to 2016.[13–15] In the same period, neonatal mortality rate (NMR) has also steadily declined (from 39 to 21 per 1000 LBs).[13–15] Being an important determinant of neonatal mortality, PTB has become a greater contributor to under-5 mortality over time.[16] If we do not consider interventions to address PTBs, it would be difficult to achieve Nepal's Sustainable Development Goal that aims to reduce the neonatal mortality to 12 per 1000 LBs and under-5 mortality to 28 per 1000 LBs by 2030.[17]

There are very few studies on the prevalence or risk factors for PTB in Nepal,[18 19] and those that exist have limitations. First, those studies are hospital based. Women enrolled in hospitals during delivery may suffer from systematic recall bias, where women having a PTB might report differently from women with term births. Also, at the time of delivery, women might have recall issues in reporting their date of LMP. Most importantly, enrolling at facilities has a selection bias, where the PTBs delivered at home or on the way to facilities are missed, possibly leading to underestimation of the prevalence and a different distribution of risk factors. Second, previous studies have included deliveries taken from urban tertiary hospitals in Nepal. Around 80% of the Nepalese population resides in rural areas[20] and does not have access to delivery services at tertiary centres. Moreover, in rural areas, only 47% of deliveries are assisted by skilled birth attendants.[14] So, the findings from those studies may not be representative of rural Nepal. Third, since the women's enrolment was during delivery, they looked at only risk factors that did not vary in pregnancy and did not analyse changing symptoms, behaviours and maternal weight gain throughout pregnancy. Some of these symptoms may be indicative of conditions that can be addressed by antenatal care (ANC). The objective of our study was to estimate the prevalence and identify predictors/risk factors of spontaneous PTBs in rural Nepal. Understanding and addressing such risk factors are critical to addressing neonatal and child mortality and morbidity, particularly in resource-poor settings like Nepal.

## METHODS
### Study design
This is a secondary data analysis with data taken from the Nepal Oil Massage Study (NOMS), which is a cluster-randomised community-based trial (ClinicalTrials.gov, NCT01177111) on the impact of sunflower seed oil versus standard of care mustard seed oil for neonatal massage on neonatal mortality and morbidity in rural Sarlahi district of Nepal. This study began by identifying married women of childbearing age (15–40 years) who consented to pregnancy surveillance. This involved following them every 5 weeks to see whether they became pregnant, based on a positive pregnancy test offered by the study team if a woman reported missing a period. If pregnant, they were consented and enrolled in the trial. During enrolment, demographic data, socioeconomic status, reproductive history and date of last menstruation were collected. One hundred twenty-three women (0.3%) refused to be followed after enrolment. Those who consented were visited monthly by a field worker until the pregnancy outcome occurred or the study ended. During these monthly visits, field workers asked some basic questions about signs and symptoms of morbidity during the previous 30-day period. At these visits, women also had their weight and blood pressure (BP)/pulse measured, and body temperature recorded. Women reporting signs of morbidity and indicating that these signs were currently present were referred to the local health post or primary health centre. Women with fever or elevated BP as measured by study staff were similarly referred for care but continued to be included in the study.

As soon as possible after labour began or the baby was delivered, family members or neighbours notified the local female study worker of the birth. She notified a specially trained team who visited the mother and infant as soon after birth as possible. They measured infant weight and time of weight measurement after birth, determined sex of the newborn and whether the baby was a singleton or multiple births.

### Setting and participants
The study cohort consists of 40119 pregnancies among married women of childbearing age, living in 34 Village Development Committees of Sarlahi district, enrolled from 9 September 2010 to 16 January 2017, in the NOMS. Pregnancies were followed monthly until delivery. LBs were categorised as term or preterm. Pregnancies ending in miscarriage, abortion and stillbirths (SBs) were excluded from the analysis. SBs were not included because the aetiology of these may be quite different from those of PTBs.

### Variables
#### Outcome variable
The main outcome variable is spontaneous PTB among pregnancies that produced at least one live born infant, defined as pregnancies ending less than 259 gestational days from the first day of LMP date. LBs were based on

women's self-report. They were asked if the baby moved, cried or breathed after birth. If they said 'yes' to one or more of these, the birth was recorded as an LB. For gestational age (GA), women were asked about their LMP during enrolment, and the GA at outcome was calculated as the difference between reported LMP and the date of the child's birth. PTBs were classified as spontaneous or non-spontaneous (caesarean section or/and induction), and only spontaneous PTBs were included in the regression analysis.

## Independent variables

Through literature review and expert opinion, certain factors were included in the analysis of predictors.[21] These can be categorised into pregnancy non-varying and pregnancy-varying variables. Pregnancy non-varying variables included sociodemographic, pregnancy history, current pregnancy and child-related variables that do not change during pregnancy. Pregnancy-varying variables included signs and symptoms of morbidity in pregnancy and maternal weight.

Sociodemographic variables like maternal age at LMP, caste/religion, maternal education, wealth quintile and maternal height were explored. Maternal age was categorised as less than 18, 18–35 and more than 35 years to assess the association of very young women and older women with PTBs. Caste/religion of the mothers (Brahmin/Chhetri, Vaishya, Shudra, Muslim and others) was used as per the caste category system in Nepal.[22] Maternal education (no schooling, 1–5 years and more than 5 years) and maternal height (<145 cm, 145–<150 and ≥150) were used. Household wealth status was measured in quintiles based on a standardised score using principal component analysis of household assets.[23]

Prior pregnancy-related variables like parity (one to four, more than four, prior pregnancy but not resulting in LB or SB and no prior pregnancy); interpregnancy interval (IPI) defined as the time since the end of last pregnancy to the date of LMP of the current pregnancy, regardless of the outcome (<18, 18–36 and >36 months); any prior live born child who died (no prior LB died and died); any prior pregnancy that ended in an SB (no prior SB and SB); any prior pregnancy ending in miscarriage (no prior miscarriage and miscarriage); and any prior pregnancy that ended in multiple births (no prior multiple births and multiple births) were assessed.

Current pregnancy-related variables like tobacco intake (ever used any tobacco products during this pregnancy—yes and no) and alcohol intake (ever used alcohol during this pregnancy—yes and no) were assessed. Child-level variables like multiple births (singleton and twin/triplet) and sex of the child (male and female) were included. We used the category with the low risk according to literature of similar settings, to be the reference group if there was no clear hierarchy of risk (such as maternal age, caste) but selected the most at risk group for those where a hierarchy existed (such as maternal education, wealth quintile, maternal height).

Current pregnancy-related variables like tobacco and alcohol intake were not included in the regressions because rates of use were very low. Only 0.3% consumed alcohol and only 1.1% used tobacco. Other current pregnancy-related variables like number of ANC visits and place of delivery were shown in descriptive, but omitted from inferential analysis because in this setting, women with spontaneous PTBs could have missed the fourth ANC visit in the ninth month and preterm birth could be the cause of a lower number of visits. For place of delivery, spontaneous PTBs were more likely to be delivered at home or on the way to the facility, because many births in this environment are not planned to occur in a facility. However, we also included these variables in the multivariable regressions and provided these as supplemental analyses because ANC may be important in reducing PTB.

Symptoms of morbidity during pregnancy such as sexually transmitted infections (STIs), respiratory illness, gastrointestinal (GI) illness, poor appetite, nausea and vomiting, vaginal bleeding, swelling of hands or face, high systolic and diastolic BP were assessed. All these variables were assessed in the second and third trimesters, and so labelled as: problem in at least one visit of the second trimester—yes or no, and problem in at least one visit in the third trimester—yes or no. We did not include symptoms of morbidities in the first trimester because only 41% women were enrolled in the first trimester, and so 59% missed symptom information in the first trimester. Maternal weight gain was defined as the average weight in the third trimester minus the average weight in the second trimester. For measurement of these symptom variables, field workers asked if women had symptoms of morbidity at any time in the past 30 days, at each monthly visit during pregnancy. STI was defined as painful or burning urination, or foul-smelling vaginal discharge. Respiratory illness was defined as persistent cough, or difficult or rapid breathing, or wheezing/grunting, or shortness of breath. GI illness was defined as watery stools (four or more times in a day or blood or white mucus in the stool). Appetite-related illness was defined as poor appetite, nausea or vomiting. Vaginal bleeding was defined as spots of blood from the vagina. Swelling was defined as swelling of hands and/or face. Foot/leg swelling was excluded since it is common during pregnancy and not indicative of underlying disease. BP measurements were categorised as high systolic BP if the systolic measurement was ≥140 mm Hg, and high diastolic BP if diastolic measurement was ≥90 mm Hg at any monthly visit within the second or third trimester.

## Statistical methods

First, a descriptive analysis was done to show the frequencies of pregnancy non-varying variables (sociodemographic, prior pregnancy related, current pregnancy and child related) and pregnancy-varying variables (symptoms and maternal weight) by spontaneous preterm and term births. Second, bivariable Generalized Estimating Equations (GEE) Poisson regression with robust variance

was used to examine associations between each risk factor and the outcome to get an unadjusted risk ratio. Since the prevalence of our outcome was more than 10%, we used Poisson regression with robust variance because we wanted to report associations as risk ratios. Third, multi-variable GEE Poisson regression with robust variance was used including variables that were significant in the bivari-able models, to get the adjusted risk ratios (ARRs). GEE was used because in the study, 52% of women had multiple pregnancies. Since our unit of analysis is pregnancy and pregnancies were nested within women, women's ID vari-able was used as cluster for GEE modelling.

We included a larger number of potential risk factors to provide a general description of the study population but did not include all of these in the regression analysis. Some variables were highly correlated with each other (such as some reproductive history variables) and we chose just to include one rather than all, and for others, the prevalence was so low that we did not think helpful to include in the regression (for example, smoking and alcohol use). Some of the variables in the unadjusted analysis were not included in the regression because they were not statistically significant in the unadjusted analysis. For example, prior pregnancy ending in miscarriage, SB or prior multiple births were not included (as these were highly correlated with each other and not statistically significant in crude models). We did include death of a prior LB, which was significant in the crude model.

The descriptive analysis had 31 851 pregnancies. In the regression analysis, we excluded the 1093 pregnan-cies (3.4%) that ended in caesarean section, induction or both, which leaves 30 758 for analysis. Then, 30.7% out of 30 758 (20.2% missing morbidity in second trimester due to enrolment only in third trimester, 9.4% missing morbidity in third trimester and 1.1% missing other vari-ables) were missing in the regression analysis, and so the final multivariable regression analysis excluded those 9461 pregnancies, and consisted of 21 297 pregnancies.

### Patient and public involvement
Patients or the public were not involved in the design, or conduct, or reporting, or dissemination plans of this study.

## RESULTS
### Participants
The analytical population is 31 851 pregnancies that ended in at least one LB and had information on GA at outcome. The detailed flow chart is given in figure 1. Most women were enrolled in the first and second trimes-ters (41% each), followed by the third trimester (18%). Overall, the mean GA at enrolment was 18 weeks. For first, second and third trimesters, the mean GAs at enrol-ment were 9, 19 and 34 weeks, respectively. Fifty-two per cent of women (33% with two pregnancies, 14% with three pregnancies, 4% with four pregnancies and 1%

with more than four pregnancies) contributed more than one pregnancy to the study.

### Descriptive analysis
For pregnancy non-varying variables, as seen in table 1, 15% of women were younger (less than 18 years) and 2% of women were older (more than 35 years of age). Nine per cent of women were Muslim caste/religion. Two-thirds of women did not go to school, whereas only nearly one-fourth had education of more than 5 years. Fifteen per cent of women had height <145 cm. About one-third (29%) of women had their first pregnancy in this study and 64% had one to four prior LBs or SBs. Among those who had a previous pregnancy, 6% had prior SB, 16% experienced miscarriage and 16% had an LB that died, and only 1% had prior multiple births. Half of the women had an IPI of less than 18 months, and 28% of women had four or more ANC visits. Half of the babies were born at home and 2% were born on the way to a facility or outdoors. Only 1.1% consumed tobacco and only 0.3% consumed alcohol during pregnancy. Half of the current pregnancies (51%) resulted in male children, and less than 1% resulted in multiple births. Only 3.4% of preg-nancies underwent either caesarean section or induction or both.

For pregnancy-varying variables, as seen in table 2, poor appetite, nausea and vomiting was the most commonly reported symptom in both the second (39%) and third trimesters (20%); and vaginal bleeding was the least reported symptom (1.2% in the second and 0.6% in the third trimester). Very few women had high systolic BP (0.5% and 0.8%) and high diastolic BP (1.5% and 2.9%) in second and third trimesters, respectively. The average weight gained by women from second to third trimester was 3.5 kg.

### Outcome data
There were 4792 PTBs out of 31 851 pregnancies with at least one LB. Hence, the prevalence of PTB was 15% (95% CI: 14.6% to 15.4%) among the pregnancies enrolled between 9 September 2010 and 16 January 2017. Spontaneous PTB was 14.5% and non-spontaneous PTB was 0.5%. On looking at severity of spontaneous PTB, the prevalence were 0.5%, 1.4%, 2.1% and 10.5% for extreme PTB (<28 weeks), very PTB (28–<32 weeks), moderate PTB (32–<34 weeks) and late PTB (34–<37 weeks), respectively.

### Main results
The main results are shown in table 3. Pregnancy non-varying variables that increased the risk of spontaneous PTB were maternal age less than 18 years (ARR=1.13, 95% CI: 1.02 to 1.26); being Muslim compared with Brahmin and Chhetri (1.53, 1.16 to 2.01); first pregnancy as compared with parity 1–4 (1.15, 1.04 to 1.28); having multiple births (4.91, 4.20 to 5.75) and having a male child (1.10, 1.02 to 1.17). Pregnancy non-varying vari-ables that decreased the risk of spontaneous PTB were

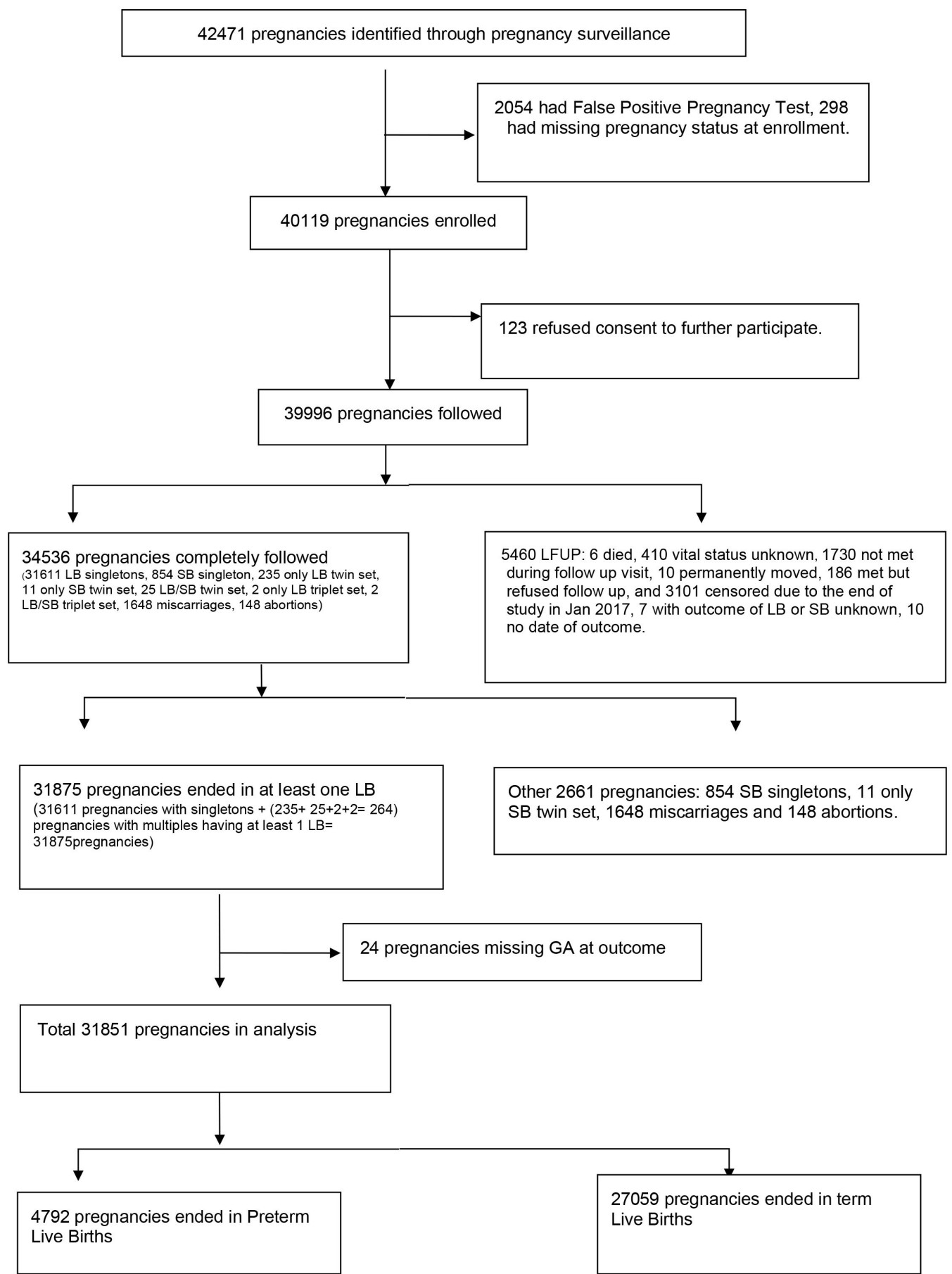

**Figure 1** Flow diagram of participants. GA, gestational age; LB; live birth; LFUP, loss to follow-up; SB, stillbirth.

**Table 1** Distribution of pregnancy non-varying variables by preterm and term births

| Variables | Categories | Total N=31 851 N (%) | Term N=27 059 N (%) | Preterm N=4792 N (%) |
|---|---|---|---|---|
| Maternal age at LMP | 18–35 | 26 206 (82.3) | 22 423 (82.9) | 3783 (78.9) |
| | Less than 18 | 4946 (15.5) | 4100 (15.2) | 846 (17.7) |
| | More than 35 | 699 2.2) | 536 2.0) | 163 (3.4) |
| Caste/religion | Brahmin and Chhetri | 963 (3.0) | 857 (3.2) | 106 (2.2) |
| | Vaishya | 22 946 (72.0) | 19 701 (72.8) | 3245 (67.7) |
| | Shudra | 4922 (15.5) | 4111 (15.2) | 811 (16.9) |
| | Muslim and others | 2989 (9.4) | 2365 (8.7) | 624 (13.0) |
| | Missing | 31 (0.1) | 25 (0.1) | 6 (0.1) |
| Maternal education | No schooling | 21 427 (67.3) | 17 915 (66.2) | 3512 (73.3) |
| | 1–5 years | 2713 (8.5) | 2330 (8.6) | 383 (8.0) |
| | More than 5 years | 7681 (24.1) | 6786 (25.1) | 895 (18.7) |
| | Missing | 30 (0.1) | 28 (0.1) | 2 (0.0) |
| Quintiles of wealth | Poorest | 6510 (20.4) | 5340 (19.7) | 1170 (24.4) |
| | Poor | 6380 (20.0) | 5403 (20.0) | 977 (20.4) |
| | Middle | 6320 (19.8) | 5314 (19.6) | 1006 (21.0) |
| | Richer | 6296 (19.8) | 5470 (20.2) | 826 (17.2) |
| | Richest | 6324 (19.9) | 5516 (20.4) | 808 (16.9) |
| | Missing | 21 (0.1) | 16 (0.1) | 5 (0.1) |
| Maternal height (cm) | <145 | 4689 (14.7) | 3885 (14.4) | 800 (16.7) |
| | 145–<150 | 9559 (29.9) | 8025 (29.7) | 1527 (31.9) |
| | ≥150 | 17 581 (55.1) | 15 111 (55.8) | 2454 (51.2) |
| | Missing | 51 (0.2) | 38 (0.14) | 11 (0.2) |
| Parity including both LB and SB, at enrolment | Parity 1–4 | 20 317 (63.8) | 17 366 (64.2) | 2951 (61.6) |
| | More than 4 | 1383 (4.3) | 1117 (4.1) | 266 (5.6) |
| | Prior pregnant but parity 0 | 787 (2.5) | 672 (2.5) | 115 (2.4) |
| | No prior pregnancy | 9195 (28.9) | 7769 (28.7) | 1426 (29.8) |
| | Missing | 169 (0.5) | 135 (0.5) | 34 (0.7) |
| Interpregnancy interval based on maternal recall | 18–36 months | 7927 (24.9) | 6787 (25.1) | 1140 (23.8) |
| | Less than 18 months | 11 461 (36.0) | 9701 (35.9) | 1760 (36.7) |
| | More than 36 months | 3256 (10.2) | 2794 (10.3) | 462 (9.6) |
| | No prior pregnancy | 9195 (28.9) | 7769 (28.7) | 1426 (29.8) |
| | Missing | 12 (0.0) | 8 (0.0) | 4 (0.1) |
| Any deaths among prior LB | Prior LB but not died | 17 488 (54.9) | 14 999 (55.4) | 2489 (51.9) |
| | Prior LB died | 3618 (11.4) | 2999 (11.1) | 619 (12.9) |
| | Prior pregnancy but no LB | 1073 (3.4) | 909 (3.4) | 164 (3.4) |
| | No prior pregnancy | 9195 (28.9) | 7769 (28.7) | 1426 (29.8) |
| | Missing | 477 (1.5) | 383 (1.4) | 94 (2.0) |
| Any prior pregnancy ended in SB | Prior pregnancy but no SB | 21 270 (66.8) | 18 127 (67.0) | 3143 (65.6) |
| | Prior SB | 1371 (4.3) | 1150 (4.2) | 221 (4.6) |
| | No prior pregnancy | 9195 (28.9) | 7769 (28.7) | 1426 (29.8) |
| | Missing | 15 (0.0) | 13 (0.0) | 2 (0.0) |
| Any prior pregnancy ended in miscarriage | Prior pregnancy but no miscarriage | 19 025 (59.7) | 16 176 (59.8) | 2849 (59.5) |

**Table 1** Continued

| Variables | Categories | Total | Term | Preterm |
| | | N=31851 | N=27059 | N=4792 |
| | | N (%) | N (%) | N (%) |
|---|---|---|---|---|
| | Prior miscarriage | 3621 (11.4) | 3104 (11.5) | 517 (10.8) |
| | No prior pregnancy | 9195 (28.9) | 7769 (28.7) | 1426 (29.8) |
| | Missing | 10 (0.0) | 10 (0.0) | 0 (0.0) |
| Any prior pregnancy ended in multiple births | Prior pregnancy but no multiple births | 22343 (70.1) | 19030 (70.3) | 3313 (69.1) |
| | Prior multiple births | 292 (0.9) | 241 (0.9) | 51 (1.1) |
| | No prior pregnancy | 9195 (28.9) | 7769 (28.7) | 1426 (29.8) |
| | Missing | 21 (0.1) | 19 (0.1) | 2 (0.0) |
| Number of ANC visits | None | 5520 (17.3) | 4524 (16.7) | 996 (20.8) |
| | 1 | 4146 (13.0) | 3420 (12.6) | 726 (15.2) |
| | 2–3 | 9779 (30.7) | 8158 (30.1) | 1621 (33.8) |
| | 4 or more | 8909 (28.0) | 8021 (29.6) | 888 (18.5) |
| | Missing | 3497 (11.0) | 2936 (10.9) | 561 (11.7) |
| Place of delivery | Home/Maiti | 15776 (49.5) | 13270 (49.0) | 2506 (52.3) |
| | HP/clinic/hospital | 12016 (37.7) | 10406 (38.5) | 1610 (33.6) |
| | Way to facility/outdoors | 610 (1.9) | 486 (1.8) | 124 (2.6) |
| | Missing | 3449 (10.8) | 2897 (10.7) | 552 (11.5) |
| Bidi or tobacco use in pregnancy | No | 31498 (98.9) | 26789 (99.0) | 4709 (98.3) |
| | Yes | 353 (1.1) | 270 (1.0) | 83 (1.7) |
| Alcohol use (jaard or rakshi) in pregnancy? | No | 31756 (99.7) | 26982 (99.7) | 4774 (99.6) |
| | Yes | 95 (0.3) | 77 (0.3) | 18 (0.4) |
| Multiple births | Singleton | 31587 (99.2) | 26946 (99.6) | 4641 (96.8) |
| | Twin/triplet | 264 (0.8) | 113 (0.4) | 151 (3.2) |
| Sex of the child | Female | 15182 (47.7) | 13063 (48.3) | 2119 (44.2) |
| | Male | 16306 (51.2) | 13794 (51.0) | 2512 (52.4) |
| | Twin/triplet | 264 (0.8) | 113 (0.4) | 151 (3.2) |
| | Missing | 99 (0.3) | 89 (0.3) | 10 (0.2) |
| Induction or CS done | Only induction | 193 (0.6) | 166 (0.6) | 27 (0.6) |
| | Only CS | 868 (2.7) | 735 (2.8) | 133 (2.8) |
| | Both induction and CS | 32 (0.1) | 28 (0.1) | 4 (0.08) |
| | None | 30758 (96.6) | 26130 (96.6) | 4628 (96.6) |

ANC, antenatal care; CS, caesarean section; HP, Health Post; LB, live birth; LMP, last menstrual period; SB, stillbirth.

maternal education of more than 5 years (0.81, 0.73 to 0.90); maternal height of ≥150 cm (0.89, 0.81 to 0.98) and being wealthier: richer (0.83, 0.74 to 0.93) wealth quintile compared with the poorest wealth quintile. Pregnancy non-varying variables that showed no association with spontaneous PTBs in the bivariable/unadjusted models are any prior pregnancy ending in SB, any prior pregnancy ending in multiple births, any prior pregnancy ending in miscarriage and IPI. The pregnancy non-varying variable that showed an association in the bivariable model, but not in the multivariable models, was any prior pregnancy ending in death for an LB.

For morbidity symptoms, some increased the risk of preterm, and all of these showed increased risk when symptoms were present in the third trimester. Having vaginal bleeding (ARR=1.53, 95% CI: 1.08 to 2.18); swelling (1.37, 1.17 to 1.60); high systolic BP (1.47, 1.08 to 2.01) and high diastolic BP (1.41, 1.17 to 1.70) in the third trimester significantly increased the risk of spontaneous PTB. Some symptom variables significantly decreased the risk of spontaneous PTB. Having respiratory problem in the third trimester (0.86, 0.79 to 0.94); and having poor appetite, nausea and vomiting in the second trimester (0.86, 0.80 to 0.92) and in the third trimester (0.86, 0.79

**Table 2** Distribution of pregnancy-varying variables by preterm and term births

| Variables | | Total N=31 851 N (%) | Term N=27 059 N (%) | Preterm N=4792 N (%) |
|---|---|---|---|---|
| STI in at least one visit of 2nd trimester? | No | 20 823 (65.4) | 17 497 (64.7) | 3326 (69.4) |
| | Yes | 4593 (14.4) | 3855 (14.2) | 738 (15.4) |
| | Missing | 6435 (20.2) | 5707 (21.1) | 728 (15.2) |
| STI in at least one visit of 3rd trimester? | No | 25 931 (81.4) | 22 512 (83.2) | 3419 (71.3) |
| | Yes | 2963 (9.3) | 2569 (9.5) | 394 (8.2) |
| | Missing | 2957 (9.3) | 1978 (7.3) | 979 (20.4) |
| Respiratory problems in at least one visit of 2nd trimester? | No | 17 963 (56.4) | 15 081 (55.7) | 2882 (60.1) |
| | Yes | 7452 (23.4) | 6271 (23.2) | 1181 (24.6) |
| | Missing | 6436 (20.2) | 5707 (21.1) | 729 (15.2) |
| Respiratory problems in at least one visit of 3rd trimester? | No | 22 860 (71.8) | 19 743 (73.0) | 3117 (65.0) |
| | Yes | 6034 (18.9) | 5338 (19.7) | 696 (14.5) |
| | Missing | 2,957 (9.3) | 1978 (7.3) | 979 (20.4) |
| GI problems in at least one visit of 2nd trimester? | No | 22 742 (71.4) | 19 136 (70.7) | 3606 (75.3) |
| | Yes | 2673 (8.4) | 2216 (8.2) | 457 (9.5) |
| | Missing | 6436 (20.2) | 5707 (21.1) | 729 (15.2) |
| GI problems in at least one visit of 3rd trimester? | No | 26 152 (82.1) | 22 712 (83.9) | 3440 (71.8) |
| | Yes | 2742 (8.6) | 2369 (8.8) | 373 (7.8) |
| | Missing | 2957 (9.3) | 1978 (7.3) | 979 (20.4) |
| Poor appetite, nausea and vomiting in at least one visit of 2nd trimester? | No | 13 121 (41.2) | 10 814 (40.0) | 2307 (48.1) |
| | Yes | 12 295 (38.6) | 10 538 (38.9) | 1757 (36.7) |
| | Missing | 6435 (20.2) | 5707 (21.1) | 728 (15.2) |
| Poor appetite, nausea and vomiting in at least one visit of 3rd trimester? | No | 22 486 (70.6) | 19 437 (71.8) | 3049 (63.6) |
| | Yes | 6409 (20.1) | 5645 (20.9) | 764 (15.9) |
| | Missing | 2956 (9.3) | 1977 (.3) | 979 (20.4) |
| Vaginal bleeding in at least one visit of 2nd trimester? | No | 25 042 (78.6) | 21 036 (77.7) | 4006 (83.6) |
| | Yes | 373 (1.2) | 315 (1.2) | 58 (1.2) |
| | Missing | 6436 (20.2) | 5708 (21.1) | 728 (15.2) |
| Vaginal bleeding in at least one visit of 3rd trimester? | No | 28 716 (90.2) | 24 938 (92.2) | 3778 (78.8) |
| | Yes | 178 (0.6) | 143 (0.5) | 35 (0.7) |
| | Missing | 2957 (9.3) | 1978 (7.3) | 979 (20.4) |
| Swelling in at least one visit of 2nd trimester? | No | 24 846 (78.0) | 20 904 (77.3) | 3942 (82.3) |
| | Yes | 571 (1.8) | 448 (1.7) | 123 (2.6) |
| | Missing | 6434 (20.2) | 5707 (21.1) | 727 (15.2) |
| Swelling in at least one visit of 3rd trimester? | No | 27 754 (87.1) | 24 126 (89.2) | 3628 (75.7) |
| | Yes | 1141 (3.6) | 956 (3.5) | 185 (3.9) |
| | Missing | 2956 (9.3) | 1977 (7.3) | 979 (20.4) |
| High systolic BP in 2nd trimester? | Normal systolic BP | 25 260 (79.3) | 21 217 (78.4) | 4043 (84.4) |
| | High systolic BP | 158 (0.5) | 136 (0.5) | 22 (0.5) |
| | Missing | 6433 (20.2) | 5706 (21.1) | 727 (15.2) |
| High systolic BP in 3rd trimester? | Normal systolic BP | 28 659 (90.0) | 24 905 (92.0) | 3754 (78.3) |

**Table 2**  Continued

| Variables | | Total N=31 851 N (%) | Term N=27 059 N (%) | Preterm N=4792 N (%) |
|---|---|---|---|---|
| | High systolic BP | 241 (0.8) | 181 (0.7) | 60 (1.3) |
| | Missing | 2951 (9.3) | 1973 (7.3) | 978 (20.4) |
| High diastolic BP in 2nd trimester? | Normal diastolic BP | 24 945 (78.3) | 20 976 (77.5) | 3969 (82.8) |
| | High diastolic BP | 473 (1.5) | 377 (1.4) | 96 (2.0) |
| | Missing | 6433 (20.2) | 5706 (21.1) | 727 (15.2) |
| High diastolic BP in 3rd trimester? | Normal diastolic BP | 27 982 (87.9) | 24 360 (90.0) | 3622 (75.6) |
| | High diastolic BP | 918 (2.9) | 726 (2.7) | 192 (4.0) |
| | Missing | 2951 (9.3) | 1973 (7.3) | 978 (20.4) |
| Average weight in 3rd trimester minus average weight in 2nd trimester in kg (mean (SD)) | | 3.5 (2.1) | 3.6 (2.1) | 2.9 (2.2) |

BP, blood pressure; GI, gastrointestinal; STI, sexually transmitted infection.

to 0.94) decreased the risk of spontaneous PTB. Symptom variables that showed no association with spontaneous PTB were STI and GI problems. Symptom variables that were significant in the bivariable model, but not significant in the multivariable models, were swelling in the second trimester and diastolic BP in the second trimester. For maternal weight, higher weight gain from the second to the third trimester was associated with a decreased risk of spontaneous PTB (0.89, 0.87 to 0.90).

To examine the possible bias associated with exclusion of pregnancies with missing data, we compared characteristics of women excluded in the regression analysis (n=9461) (mainly because of missing morbidity in second trimester due to late enrolment) with those included in the regression analysis (n=21 297) (online supplemental table 1). The women excluded in the regression analysis were slightly better off than those included in the regression based on education and socioeconomic status but most relevant, the spontaneous PTB prevalence was 17.9% for those excluded in the regression compared with 13.8% included in the regression.

We also reran the regression model including number of ANC visits. The fewer the number of ANC visits, the higher the risk of spontaneous PTB (online supplemental table 2). The other regression coefficients did not change in any qualitative way. This could be due to fewer ANC visits putting women at higher risk of spontaneous PTB as services provided in ANC (counselling, iron folic acid tablets, BP and weight measurements) are provided less often, but this association may also be due to shorter duration of pregnancy leading to less time available for ANC visits.

## DISCUSSION

Our study is one of the only large-scale studies on PTBs using data from an existing pregnancy surveillance in rural Sarlahi, Nepal. The prevalence of PTB is 15%, higher than previous estimates from Nepal,[18 19] which were primarily from urban areas and large hospital-based studies. Our study's strength is that it was population based and included all home and facility deliveries but is confined to a rural and relatively small geographical area (one-third of a district). Our study population is not necessarily representative of all of Nepal, but it is representative of Province 2 in the Terai region within which Sarlahi district is located. For example, the NMR in our study was 31 per 1000 LBs. This is similar to the NMR in the 2016 Nepal Demographic Health Survey (NDHS) for Province 2 (30 per 1000). Similarly, 67% of women in our study had no schooling, slightly higher than the 61% in the NDHS for Province 2. NDHS did not provide data on ANC 4+ for Province 2 but rural areas of Nepal had 62% coverage of ANC 4+. It should be noted in our study that healthcare seeking in pregnancy is low considering the low rates of four or more ANC visits (28%) and facility deliveries (38%). The low rates of induction and caesarean section point to a very low proportion of the PTBs being due to iatrogenic causes.

In many other settings, both younger and older maternal age have been reported to be risk factors for PTB.[24–30] Being from Muslim caste was positively associated with preterm as compared with Brahmin/Chhetri, which constitutes the major caste in Nepal. Caste/religion is a social construction, and studies in different places have shown that women in minor caste/race/colour have higher risk of PTBs.[31–33] It significantly matters what position an individual holds within a society, with regard to occurrence of diseases and also their unequal distribution.[34–36] First pregnancy (primipara) has been shown to be associated with spontaneous PTB in other studies. A study in France showed that primipara as compared with parity 2–3 increased the risk of PTB by 1.8 times.[37] Another study in the USA showed that being primipara as compared with multipara increased the risk

**Table 3** Crude and adjusted risk ratios for associations between risk factors and spontaneous preterm birth

| Name of variables | Categories | Unadjusted model | Adjusted model (N=21 297) |
| --- | --- | --- | --- |
| | | Risk ratio (95% CI) | Risk ratio (95% CI) |
| Maternal age at LMP | 18–35 | 1 | 1 |
| | Less than 18 | 1.19*** (1.11 to 1.28) | 1.13* (1.02 to 1.26) |
| | More than 35 | 1.57*** (1.36 to 1.81) | 1.22 (0.98 to 1.51) |
| Caste/religion categories | Brahmin and Chhetri | 1 | 1 |
| | Vaishya | 1.33** (1.09 to 1.62) | 1.23 (0.95 to 1.59) |
| | Shudra | 1.55*** (1.26 to 1.90) | 1.23 (0.94 to 1.62) |
| | Muslim and others | 1.96*** (1.60 to 2.42) | 1.53** (1.16 to 2.01) |
| Mother's years of education | No schooling | 1 | 1 |
| | 1–5 years | 0.86** (0.78 to 0.95) | 0.91 (0.80 to 1.03) |
| | More than 5 years | 0.71*** (0.66 to 0.76) | 0.81*** (0.73 to 0.90) |
| Quintiles of wealth | Poorest | 1 | 1 |
| | Poor | 0.86*** (0.79 to 0.93) | 0.90* (0.82 to 1.00) |
| | Middle | 0.89** (0.82 to 0.96) | 0.95 (0.86 to 1.05) |
| | Richer | 0.73*** (0.67 to 0.79) | 0.83** (0.74 to 0.93) |
| | Richest | 0.71*** (0.65 to 0.77) | 0.88* (0.78 to 1.00) |
| Mother's height (cm) | <145 | 1 | 1 |
| | 145–<150 | 0.93 (0.86 to 1.01) | 0.98 (0.88 to 1.08) |
| | ≥150 | 0.81*** (0.75 to 0.87) | 0.89* (0.81 to 0.98) |
| Parity including both LB and SB, at enrolment | Parity 1–4 | 1 | 1 |
| | More than 4 | 1.32*** (1.17 to 1.48) | 1.17 (0.99 to 1.37) |
| | Prior pregnancy but parity 0 | 1.02 (0.85 to 1.22) | 0.92 (0.62 to 1.37) |
| | No prior pregnancy | 1.10** (1.04 to 1.17) | 1.15** (1.04 to 1.28) |
| Interpregnancy interval | 18–36 months | 1 | 1 |
| | Less than 18 months | 1.07 (0.99 to 1.14) | 1.08 (0.99 to 1.18) |
| | More than 36 months | 0.98 (0.89 to 1.09) | 0.9 (0.79 to 1.02) |
| | No prior pregnancy | 1.11** (1.03 to 1.20) | 1 (1.00 to 1.00) |
| Any death among prior LB | Prior LB but not died | 1 | 1 |
| | Prior LB died | 1.19*** (1.09 to 1.29) | 1.07 (0.97 to 1.19) |
| | Prior pregnancy but no LB | 1.07 (0.92 to 1.25) | 1.06 (0.75 to 1.49) |
| | No prior pregnancy | 1.12*** (1.06 to 1.19) | 1 (1.00 to 1.00) |
| Any prior pregnancy ended in SB | Prior pregnancy but no SB | 1 | |
| | Prior SB | 1.08 (0.94 to 1.23) | |
| | No prior pregnancy | 1.08** (1.02 to 1.15) | |
| Any prior pregnancy ended in miscarriage | Prior pregnancy but no miscarriage | 1 | |
| | Prior miscarriage | 0.94 (0.86 to 1.03) | |
| | No prior pregnancy | 1.07* (1.01 to 1.13) | |
| Any prior pregnancy ended in multiple births | Prior pregnancy but no multiple births | 1 | |
| | Prior multiple births | 1.14 (0.87 to 1.49) | |
| | No prior pregnancy | 1.08** (1.02 to 1.14) | |
| Multiple births | Singleton | 1 | 1 |
| | Twin/triplet | 3.92*** (3.52 to 4.38) | 4.91*** (4.20 to 5.75) |
| Sex of the child | Female | 1 | 1 |
| | Male | 1.10*** (1.04 to 1.17) | 1.10** (1.02 to 1.17) |

Continued

**Table 3** Continued

| Name of variables | Categories | Unadjusted model Risk ratio (95% CI) | Adjusted model (N=21 297) Risk ratio (95% CI) |
|---|---|---|---|
| | Twin/triplet | 4.13*** (3.69 to 4.63) | 1 |
| STI in at least one visit of 2nd trimester? | No | 1 | |
| | Yes | 0.99 (0.92 to 1.07) | |
| STI in at least one visit of 3rd trimester? | No | 1 | |
| | Yes | 1.01 (0.92 to 1.12) | |
| Respiratory problems in at least one visit of 2nd trimester? | No | 1 | 1 |
| | Yes | 1 (0.94 to 1.06) | 1.08 (1.00 to 1.16) |
| Respiratory problems in at least one visit of 3rd trimester? | No | 1 | 1 |
| | Yes | 0.85*** (0.79 to 0.92) | 0.86** (0.79 to 0.94) |
| GI problems in at least one visit of 2nd trimester? | No | 1 | |
| | Yes | 1.08 (0.98 to 1.18) | |
| GI problems in at least one visit of 3rd trimester? | No | 1 | |
| | Yes | 1.04 (0.94 to 1.16) | |
| Poor appetite, nausea and vomiting in at least one visit of 2nd trimester? | No | 1 | 1 |
| | Yes | 0.81*** (0.77 to 0.86) | 0.86*** (0.80 to 0.92) |
| Poor appetite, nausea and vomiting in at least one visit of 3rd trimester? | No | 1 | 1 |
| | Yes | 0.88** (0.82 to 0.95) | 0.86*** (0.79 to 0.94) |
| Vaginal bleeding in at least one visit of 2nd trimester? | No | 1 | 1 |
| | Yes | 0.91 (0.71 to 1.17) | 0.84 (0.71 to 1.17) |
| Vaginal bleeding in at least one visit of 3rd trimester? | No | 1 | 1 |
| | Yes | 1.44* (1.05 to 1.98) | 1.53* (1.08 to 2.18) |
| Swelling in at least one visit of 2nd trimester? | No | 1 | 1 |
| | Yes | 1.32*** (1.12 to 1.55) | 1.19 (0.98 to 1.46) |
| Swelling in at least one visit of 3rd trimester? | No | 1 | 1 |
| | Yes | 1.25** (1.09 to 1.44) | 1.37*** (1.17 to 1.60) |
| High systolic BP in 2nd trimester? | Normal systolic BP | 1 | 1 |
| | High systolic BP | 0.89 (0.59 to 1.34) | 0.67 (0.40 to 1.12) |
| High systolic BP in 3rd trimester? | Normal systolic BP | 1 | 1 |
| | High systolic BP | 1.92*** (1.52 to 2.41) | 1.47* (1.08 to 2.01) |
| High diastolic BP in 2nd trimester? | Normal diastolic BP | 1 | 1 |
| | High diastolic BP | 1.34** (1.12 to 1.60) | 1.09 (0.85 to 1.40) |
| High diastolic BP in 3rd trimester? | Normal diastolic BP | 1 | 1 |
| | High diastolic BP | 1.57*** (1.37 to 1.80) | 1.41*** (1.17 to 1.70) |
| Average weight in 3rd trimester minus average weight in 2nd trimester (kg) | | 0.88*** (0.87 to 0.90) | 0.89*** (0.87 to 0.90) |

*P<0.05, **p<0.01, ***p<0.001.
BP, blood pressure; GI, gastrointestinal; LB, live birth; LMP, last menstrual period; SB, stillbirth; STI, sexually transmitted infection.

of very preterm and extremely PTB, with the highest risk of 1.37 times for extremely PTB.[38] Meta-analysis done using 14 cohort studies from low/middle-income countries (LMICs)[39] and a study from sub-Saharan African countries[40] also show that primiparity is associated with increased odds of PTB. Primipara is a risk factor for hypertensive disorders of pregnancy, which increases the risk of PTB.[41] Our study did not show IPI to be the risk

factor for spontaneous PTB. However, other studies on relationships between IPI and PTB consistently showed that shorter IPIs increase the risk of PTBs. However, the intervals used were not uniform across studies. One study found that, compared with an IPI of 18–23 months, IPIs <3, 3–5 and 6–12 months had higher risks of PTB.[42] Another study with median IPI of 36 months showed that, compared with an IPI of 24–36 months, an IPI of <24

months was associated with preterm delivery.[43] Different studies corroborate our finding that multiple births are a risk factor for PTB.[18 44 45] Similar to our study, others also found male children at higher risk of being preterm,[46–48] but a study in Nepal found that female children had a higher risk of being preterm.[18] This study in Nepal enrolled LBs in a hospital setting, and had almost half the prevalence of our study.[18] They could have missed more boys that had PTBs at home or on the way to a facility.

Different studies in Nepal[18] and outside of Nepal[49–51] have also shown that higher education of mothers decreases the risk of PTBs. Higher education of mothers can lead to increased knowledge and awareness regarding pregnancy-related care and thus decrease adverse outcomes of pregnancy. We found greater maternal height to be protective for spontaneous PTB, similar to the findings from a meta-analysis done using 12 cohort studies from LMICs.[52] We found that women in the richer wealth quintile had a lower risk of spontaneous PTBs. Having higher household economic status probably does not directly affect the GA at outcome, instead, it probably is mediated by factors like nutrition, physically demanding work during pregnancy, type of care at home, stress level and other psychological factors.[53]

Pregnancy-varying morbidities that significantly decreased the risk of PTB in our analysis were respiratory problems in the third trimester, and poor appetite, nausea and vomiting in the second trimester and the third trimester. On segregating the symptoms within respiratory problems, we found that it was the persistent cough in the third trimester that decreased the risk of PTB. A similar relationship was found between persistent cough and large for GA in another study done using the same data as ours.[54] However, we could not find any such association in the previous literature. The association might be due to some unmeasured confounders, or it could be that women with persistent cough in the third trimester made more frequent check-up visits. We saw that 40% of women with persistent cough in the third trimester sought treatment for cough, and almost all had sought treatment more than once. The pathogenesis of nausea and vomiting in pregnancy is not very clear, but it is broadly accepted to be multifactorial, with the involvement of genetic, endocrine and GI factors.[55] Our findings corroborate with previous findings that nausea and vomiting is associated with reduced risk of PTB.[56–59] Specifying by trimesters, a study by Wallin et al in Nepal showed similar findings—poor appetite, nausea and vomiting in the first trimester were not significantly associated with spontaneous PTBs, but having these symptoms in the second trimester decreased the risk of spontaneous PTB by 25%.[60]

Pregnancy-varying morbidities that significantly increased the risk of spontaneous PTB were vaginal bleeding, swelling of hands and face, high diastolic and systolic BP, all in the third trimester. Vaginal bleeding is associated with fetal exposure to oral pathogens, which thereby increases the risk of spontaneous PTB; however,

whether bleeding is the cause or result of fetal exposure to oral pathogens is not clear.[61] A prospective cohort study, separating first and second trimesters, showed that vaginal bleeding in both trimesters increased the risk of PTB by 3.6 times, while bleeding in the second trimester only, was not associated with PTB.[62] A systematic review using 23 studies showed that bleeding in early pregnancy increased the risk of PTBs.[63] A study in China showed that vaginal bleeding in the first trimester increased the risk of PTBs, and the severity, duration and initial timing of vaginal bleeding had different effects on the severity of PTBs.[61] Due to the low enrolment of women in the first trimester, we could not look at the association of vaginal bleeding in the first trimester with spontaneous PTB. However, all of the above information indicates that vaginal bleeding can be an important predictor of spontaneous PTB and healthcare workers should recommend appropriate interventions for women if they present with vaginal bleeding (such as more frequent follow-up or referral for higher level care).

Other studies on BP during pregnancy have also shown that a rise in systolic BP (over 30 mm Hg) or diastolic BP (over 15 mm Hg), from early pregnancy to the mid-third trimester, significantly increased the risk of spontaneous PTB by two to three times.[64] Another study showed that an increase in 10 mm Hg in diastolic BP increased the risk of PTB by 29%.[65] These indicate the importance of measuring BP during the third trimester. High BP in the third trimester is an indicator of pre-eclampsia/eclampsia and can predict PTB. Measuring BP frequently and monitoring the rise and cause of increased BP are important for predicting spontaneous PTB.

For maternal weight, higher weight gain from the second to the third trimester decreased the risk of spontaneous PTB. This is consistent with a study done outside Nepal, which showed that very low weight gain was strongly associated with very preterm delivery, and that this varied by pre-pregnancy body mass index (BMI), where underweight women had the highest association and very obese women had lowest association with PTB.[66] Our study was conducted is a non-obese and undernourished population. We do not have pre-pregnancy BMI, so we looked at the mean BMI in the first trimester. Though the first trimester represents less half of the pregnancies in the study, it hints at undernutrition in the population. The mean BMI was 19.1 kg/m$^2$, and 37% had BMI less than 18.5 kg/m$^2$. So, less maternal weight gain in such population can pose a risk to spontaneous PTBs. Given spontaneous PTBs have shorter gestation, the increase in weight gain will likely be less because there is less time to increase weight, especially in the third trimester, when much of the gestational weight is gained.

## Strengths and limitations

This was a large population-based study that was generally representative of the rural Terai region of Nepal. Multiple variables were collected, including socioeconomic, demographic, pregnancy history and monthly morbidity in

pregnancy that could be examined as risk factors for spontaneous PTB. Although there were some missing data in regression analyses, a comparison of those with and without missing data did not show large differences in risk factor prevalence. However, those missing data had higher prevalence of PTB. It is possible that if women with missing data were included in the regression, we may have seen stronger associations but the potential bias of these differences is unclear. GA at birth was measured using date of LMP as usually done in the LMICs rather than by ultrasound. However, as LMP was asked at enrolment, which was generally early in pregnancy, there is less recall bias than LMP recalled at delivery or late in pregnancy. Using the same method as we used to obtain LMP, Gernand *et al* found that LMP-based estimates of GA in rural Bangladesh were a mean 2.8 days longer than what was obtained on ultrasound.[67] We therefore believe that this is probably not a significant limitation. Women were followed prospectively at monthly intervals to reduce recall bias about pregnancy morbidities and symptoms. In order to reduce misclassification of SBs and LBs, women were asked whether the infant moved, breathed or cried after birth.

Some variables associated with increased risk of spontaneous PTBs in previous studies, for example, a prior pregnancy ending in a PTB, gestational diabetes, maternal anaemia and pre-pregnancy maternal nutritional status were not measured in the main trial. However, other important morbidity variables were measured and used in the analysis. Some covariates were highly correlated with each other (such as some reproductive history ones) and so, not all were included in the multivariable regression. Some covariates were not statistically significant in unadjusted analyses and there was not a compelling biological or sociological reason to include them in the adjusted model. Other important variables like smoking and alcohol, although measured, could not be included in the final regression analysis as their prevalence was very low in this population. We believe these risk factors are likely generalisable for similar populations in South Asia.

## Conclusion

PTB is a leading risk factor for neonatal and under-5 mortality and morbidity worldwide. To reduce neonatal mortality, preventing PTBs can be a vital step. Some of the risk factors from our study are amenable to antenatal interventions but many others need more understanding of the underlying causal mechanisms. Maternal education and awareness can play a role in the long term, while good-quality ANC, as suggested by the new WHO recommendation of eight contacts during pregnancy, may help reduce some PTBs. Future research should focus on basic research involving the field of 'omics' using biological samples and implementation research to improve ANC and maternal nutrition.

**Author affiliations**
[1]Department of International Health, Johns Hopkins University Bloomberg School of Public Health, Baltimore, Maryland, USA
[2]Nepal Nutrition Intervention Project Sarlahi, Kathmandu, Nepal
[3]Department of Biostatistics, Johns Hopkins University Bloomberg School of Public Health, Baltimore, Maryland, USA
[4]Department of Global Health, George Washington University School of Public Health and Health Services, Washington, District of Columbia, USA

**Acknowledgements** We would like to acknowledge study staff and all participants at Nepal Nutritional Intervention Project—Sarlahi site. We would also like to give special thanks to Dr Leah Horton at Johns Hopkins Bloomberg School of Public Health for her technical contributions during data management.

**Contributors** SS, EAH, DM, SZ, REB and JK conceptualised and designed the analysis. SS conducted the analysis and wrote the manuscript. LCM, JMT, SKK, SCL and JK were investigators in the parent trial. All authors reviewed results and analysis, discussed interpretations, and contributed to development and revision of the manuscript. SS is the guarantor and accepts full responsibility for the work and/ or the conduct of the study, had access to the data, and controlled the decision to publish.

**Funding** NOMS was supported by the National Institutes for Child Health and Development (HD060712) and the Bill & Melinda Gates Foundation (OPP1084399). The analysis for this paper was funded by the National Institutes for Child Health and Development (HD092411).

**Disclaimer** The funders did not have a role in the design of the study, the data collection, nor the analysis, interpretation and writing of the manuscript.

**Competing interests** None declared.

**Patient and public involvement** Patients and/or the public were not involved in the design, or conduct, or reporting, or dissemination plans of this research.

**Patient consent for publication** Obtained.

**Ethics approval** NOMS was approved by the Institutional Review Board (IRB) of the Johns Hopkins Bloomberg School of Public Health in the USA and by the IRB of the Institute of Medicine, Tribhuvan University, Kathmandu, Nepal. This analysis of secondary data was considered exempt by the Johns Hopkins Bloomberg School of Public Health IRB (FWA00000287). Verbal consent was obtained from women for their participation and their infants for the primary data collection.

**Provenance and peer review** Not commissioned; externally peer reviewed.

**Data availability statement** No data are available.

**ORCID iDs**
Seema Subedi http://orcid.org/0000-0002-6360-3998
Elizabeth A Hazel http://orcid.org/0000-0002-9176-3278

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
