## [Reviewer comments · BMJ Open]

ARTICLE DETAILS

TITLE (PROVISIONAL)	Prevalence and Predictors of Spontaneous Preterm Births in Nepal- Findings from a prospective, population-based pregnancy cohort in rural Nepal: A secondary data analysis
AUTHORS	Subedi, Seema; Hazel, Elizabeth; Mohan, Diwakar; Zeger, Scott; Mullany, Luke C.; Tielsch, James; Khatry, Subarna; LeClerq, Steven C.; Black, Robert E.; Katz, Joanne

VERSION 1 – REVIEW

REVIEWER	Dang, Shaonong Xi'an Jiaotong University Health Science Center
REVIEW RETURNED	20-Aug-2022

GENERAL COMMENTS	Due to different population and living context, investigation to risk factors related to preterm birth is key task for improving maternal and child health. This work focused on prevalence and Predictors of Preterm Births in rural Nepal, and there are some interesting findings. Although there was a larger sample size, some technical issues should be addressed further if a revision is invited. 1. A key issue in this study is that some participants were missing due to incomplete data. In fact, the sample sizes are different for description and regression analysis. But identification of risk factors is key task of this work, so it would be better if authors could compare the participants included and excluded in regression analysis. Because missing participants could bias the association observed. Other option is that authors just use the participants with complete data, and compare the participants included and excluded in analysis.2. Although authors mentioned some limitations such as results from hospital-based study, this work still was not a representative study for Nepal because this trial seems to be conducted in 34 VDCs of Sarlahi district. This could be another limitation. Moreover, what is VDC? There is lack of detailed statement on this trial design.3. This trial was cluster-randomized community-based trial. What is cluster? This cluster variable is used in GEE. It seems not. According to methods, GEE is used to control multiple birth, is that right?4. For risk analysis, the number of covariates used for regression analysis seems to be less than that in table 1 and 2. Why? Check pls.5. It is confusing that in table 3 some variables seem not to be analyzed in adjusted model such as any prior pregnancy ended in miscarriage. Why?
--

REVIEWER	Papadopoulou, Eleni
-----------------	---------------------

	Norwegian Inst Publ Hlth, Department of Environmental Exposures and Epidemiology
REVIEW RETURNED	26-Aug-2022

GENERAL COMMENTS	The authors have studied several factors and their association with spontaneous preterm delivery in Nepal. The data collection strategy and study design are appropriate for their research aim. The quality of data is of high quality and their findings of high importance given the context of the study. However, there are some points to be clarified. See detailed comments below: Author list: This is a large study conducted in Nepal while only 2 of the 10 co-authors, none of them first or last, have an affiliation from Nepal, together with their US affiliations. I am wondering about the efforts, if any, for fair representation and geographical distribution of the contributing authors to a study from an LMIC. I understand that there is no international consensus on the criteria of fair representation among LIC/LMIC-based scientific publications in global medical research but still some efforts can be done to support LMIC authorship representation, especially when the whole study has been conducted there. Maybe the authors can comment on that. Abstract: Page 3, lines 29-30: you are referring to the factors such as age, religion, parity, gravidity and child sex as baseline characteristics. As this is not a trial, the term “baseline” does not accurately describes these characteristics as this is a mix of socio-demographic and perinatal characteristics. Please revise this terminology throughout the manuscript. The limitations of the study summarized in page 4, might need revision. The LMP for the estimation of delivery date is the most common methodology in LMIC context. Page 5, lines 37-48: is there a goal for preterm delivery reduction rates in Nepal? Page 5, lines 51-56: you are describing two previous studies exploring a similar aim as yours in Nepal, while explaining that “most studies” are hospital based. Can you please revise this sentence and explain if both or only one of the two studies are hospital-based? Page 6, lines 5-9: Do you have information about the percentage of non-facility/non-hospital or deliveries attended by skilled health personnel in urban or rural contexts in Nepal? Page 6, lines 12-15: maybe revising to “...changing symptoms and behaviors”, as pregnancy is a period characterized also by changes in behaviors, as well. Page 6, study design: do you have information regarding the % of women who did not consent and their characteristics? Do they differ from those who consented to participate? Also for those women referred by the study staff, are they included in your study or where they excluded? Please clarify. Page 8, lines 35-42: I understand the argument of less ANC contacts if the pregnancy period is shorter, due to preterm delivery. Nevertheless, ANC is a determinant of birth outcomes. My suggestion would be to include it in the main analysis. There are studies with important findings in LMICs context, related to ANC visits (i.e Pervin J et al. BMJ Open, 2020). Page 9, lines 27-30: The information of the exclusion of medically indicated preterm birth come very late in the manuscript (Methods section). Please report the % of spontaneous and non-spontaneous preterm deliveries earlier in manuscript (abstract). It
--

	would be more accurate to refer to your outcome as “spontaneous preterm birth” throughout the manuscript. Page 9, lines 50: how many of the included infants were siblings? Page 10, lines 1-10: did you consider multiple imputation or other techniques to treat your missing information, without excluding 29% of your study population? Also did the excluded 29% differ in other characteristics compared to the included women? Since you had such close monitoring and data collection strategy with hired research staff, what was the reason for this missing information? You are reporting that “only” 41% was enrolled in the 1st trimester and so you excluded the information collected in the 1st trimester but the same percentage of women were enrolled in the 2nd trimester and you kept the information collected in the 2nd trimester. This is a bit confusing. Please clarify. Table 3: how did you decide on the reference category for each variable? Table 3: what is the unit of the last variable in the table, per kg of body weight? Discussion: Page 19, lines 15-18: this statement needs context. If excluding the missing information, 66% of the women in your study (82% enrolled at 1st or 2nd trimester) attended 2 or more ANC visits. Do you still consider this as low rate of ANC in an LMIC rural context? What is the comparison for this statement? Page 19, lines 22-34: I don’t think that the comparisons with US and French populations are adequate. Please use references of studies with similar context as yours. Page 20, lines 15-27: there were several morbidities decreasing the risk for preterm birth, while the discussion of this finding is not adequate. Could the authors consider any reasons for this observation, as for example higher frequency of ANC or care seeking for this group? Page 20, lines 50-51: again the low enrollment rate on 1st trimester is used as an argument for not checking this assumption, even though you have the data and the same enrollment rate as in 2nd trimester. Please clarify. Page 21, lines 27-29: you are claiming that this is a population of normal BMI but you have not reported that in the tables. Please include this factor in your analysis. In addition, there are more proper metrics to assess weight gain in combination with pre-pregnancy BMI, used internationally. Limitations and strengths: the use of LMP for GA estimation can produce outcome misclassification bias, if compared to ultrasound-based estimations. Have you considered how big of an issue this is for your study and how it can affect your findings? In addition, the list of factors, important for preterm delivery, that have not been included in this study is much longer than anemia and GDM, like for example maternal nutrition. Please include an adequate assessment of study limitations and how they could have affected your findings. Lastly, even though a rich dataset was collected the authors have not considered that some of the factors collected might interact with each other and different effect estimates can be produced. This is a limitation of the statistical analysis strategy but it can be easily addressed.
--	---

VERSION 1 – AUTHOR RESPONSE

Reviewer: 1

Dr. Shaonong Dang, Xi'an Jiaotong University Health Science Center

Comments to the Author:

Due to different population and living context, investigation to risk factors related to preterm birth is key task for improving maternal and child health. This work focused on prevalence and Predictors of Preterm Births in rural Nepal, and there are some interesting findings. Although there was a larger sample size, some technical issues should be addressed further if a revision is invited.

1. A key issue in this study is that some participants were missing due to incomplete data. In fact, the sample sizes are different for description and regression analysis. But identification of risk factors is key task of this work, so it would be better if authors could compare the participants included and excluded in regression analysis. Because missing participants could bias the association observed. Other option is that authors just use the participants with complete data, and compare the participants included and excluded in analysis.

In the descriptive data (Table 1), we have provided the number of records that are missing data for each of the variables in the table, a subset of which we used in the regression analysis. Most variables have only a small percentage missing data. However, sources of bias (compared to the full population) are likely to come from the other greater missingness. We have revised as below:- Revised in the Statistical Method Section: The descriptive analysis had 31,851 pregnancies. In the regression analysis, we excluded the 1093 pregnancies (3.4%) that ended in caesarian section, induction or both, which leaves 30,758 for analysis. Then, 30.7% out of 30,758 (20.2% missing morbidity in 2nd trimester due to enrollment only in 3rd trimester, 9.4% missing morbidity in 3rd trimester and 1.1% missing other variables) were missing in the regression analysis, and so the final multivariable regression analysis excluded those 9,461 pregnancies, and consisted of 21,297 pregnancies.

Added in the Results-Outcome Section: To examine possible bias associated with exclusion of pregnancies with missing data, we compared characteristics of women excluded in the regression analysis (n=9,461) (mainly because of missing morbidity in 2nd trimester due to late enrollment) with those included in the regression analysis (n=21,297) (supplementary Table S1). The women excluded in the regression analysis were slightly better off than those included in the regression based on education and socioeconomic status but most relevant, the spontaneous preterm prevalence was 17.9% for those excluded in the regression compared to 13.8% included in the regression.

Added in the Discussion Section: Although there was some missing data in regression analyses, a comparison of those with and without missing data did not show large differences in risk factor prevalence. However, those missing data had higher prevalence of preterm birth. It is possible that if women with missing data were included in the regression, we may have seen stronger associations but the potential bias of these differences is unclear.

2. Although authors mentioned some limitations such as results from hospital-based study, this work still was not a representative study for Nepal because this trial seems to be conducted in 34 VDCs of Sarlahi district. This could be another limitation. Moreover, what is VDC? There is lack of detailed statement on this trial design.

VDC is an administrative unit that stands for Village Development Committee. We have clarified this in the text now.

We have not provided a lot of detail about the trial aspects of the study design because we are not analyzing those data. Rather we are using the trial's data for observational analysis. More details of the trial design are available on clinicaltrials.gov.

Our study population is not necessarily representative of all of Nepal, but it is representative of Province 2 in the Terai region within which Sarlahi district is located. For example, the NMR in our study was 31 per 1000 live births. This is similar to the NMR in the 2016 Nepal Demographic Health

Survey (NDHS) for Province 2 (30 per 1000). Similarly, 67% of women in our study had no schooling, slightly higher than the 61% in the NDHS for Province 2. We have added this to the discussion regarding the extent to which these results are representative of other populations.

3. This trial was cluster-randomized community-based trial. What is cluster? This cluster variable is used in GEE. It seems not. According to methods, GEE is used to control multiple birth, is that right?

The cluster in the trial was a geographic area which was artificially created, and contained one female data collector who monitored that area. Roughly 100 pregnant women resided in each randomized area. The female worker provided the intervention at that level, that is (either sunflower oil or mustard oil) and monitored compliance. Since we were not analyzing the results of the randomized trial, we did not account for the randomized clusters using GEE.

However, we did use in the GEE to account for the fact that an individual woman in the study may have contributed more than one pregnancies to the study. We added this - GEE was used because in the study, 52% women had multiple pregnancies. Since our unit of analysis is pregnancy and pregnancies were nested within women, women's id variable was used as cluster for GEE modelling.

4. For risk analysis, the number of covariates used for regression analysis seems to be less than that in table 1 and 2. Why? Check pls.

We have explained in the Method-Variables section that that some variables were shown to provide a general description of the study population, and not for regression analysis. Some variables were highly correlated with each other (such as some reproductive history variables) and we chose just to include one rather than all, and for others, the prevalence was so low that we did not think helpful to include in the regression (for example, smoking and alcohol use). This has been added to the methods section.

5. It is confusing that in table 3 some variables seem not to be analyzed in adjusted model such as any prior pregnancy ended in miscarriage. Why?

As explained in the methods section, some of the variables in the unadjusted analysis were not included in the regression because they were not statistically significant in the unadjusted analysis. As the reviewer mentions, prior pregnancy ending in miscarriage, stillbirth or a prior multiple birth were not included (as these were highly correlated with each other and not statistically significant in crude models). We did include death of a prior livebirth, which was significant in the crude model. This has been added to the methods section.

Reviewer: 2

Dr. Eleni Papadopoulou, Norwegian Inst Publ Hlth

Comments to the Author:

The authors have studied several factors and their association with spontaneous preterm delivery in Nepal. The data collection strategy and study design are appropriate for their research aim. The quality of data is of high quality and their findings of high importance given the context of the study. However, there are some points to be clarified. See detailed comments below:

Author list: This is a large study conducted in Nepal while only 2 of the 10 co-authors, none of them first or last, have an affiliation from Nepal, together with their US affiliations. I am wondering about the efforts, if any, for fair representation and geographical distribution of the contributing authors to a study from an LMIC. I understand that there is no international consensus on the criteria of fair representation among LIC/LMIC-based scientific publications in global medical research but still some efforts can be done to support LMIC authorship representation, especially when the whole study has been conducted there. Maybe the authors can comment on that. Let's respond that we appreciate them mentioning this issue. We think it an important one to be addressing. But please note that Ms. Subedi who is the first author, is Nepali and residing in Nepal, while working with Johns Hopkins.

Thank you for raising this important issue and we are also concerned about it. I would like to make clear that I, Seema Subedi, am from Nepal, and I work for Hopkins, but I am stationed here in Nepal. I

did my MPH from Hopkins and now work from Nepal for Hopkins. I have added my affiliation with the NNIPS organization as well now.

Abstract:

Page 3, lines 29-30: you are referring to the factors such as age, religion, parity, gravidity and child sex as baseline characteristics. As this is not a trial, the term “baseline” does not accurately describes these characteristics as this is a mix of socio-demographic and perinatal characteristics. Please revise this terminology throughout the manuscript. This is easy to address. Just change baseline to characteristics of women and infant that are not time varying (sociodemographic, pregnancy history and sex of infant) and we have conceptually separated these from ones that are time varying, such as morbidity in pregnancy.

We have removed the term “baseline” throughout the text, to describe risk factors that do not vary in pregnancy (such as socioeconomic, demographic, and pregnancy history) and term it as pregnancy non-varying, and refer now to morbidity in pregnancy as pregnancy-varying risk factors to distinguish them from those that do not vary within pregnancy.

The limitations of the study summarized in page 4, might need revision. The LMP for the estimation of delivery date is the most common methodology in LMIC context.

Thank You for noting this. We have revised the limitations in the Strength and Limitation section of the discussion. “Gestational age (GA) at birth was measured using date of last menstrual period (LMP) as usually done in the LMICs rather than by ultrasound.”

Page 5, lines 37-48: is there a goal for preterm delivery reduction rates in Nepal?

Unfortunately, the government does not have a specific goal for reduction of preterm delivery, and these are not provided in the Sustainable Development Goals either.

Page 5, lines 51-56: you are describing two previous studies exploring a similar aim as yours in Nepal, while explaining that “most studies” are hospital based. Can you please revise this sentence and explain if both or only one of the two studies are hospital-based?

We have revised to clarify that both are hospital based as “those studies are hospital-based.”

Page 6, lines 5-9: Do you have information about the percentage of non-facility/non-hospital or deliveries attended by skilled health personnel in urban or rural contexts in Nepal?

We have added the sentence – “Moreover, in rural area, only 47% of deliveries are assisted by skilled birth attendants.” We have also added in the discussion “It should also be noted in our study that health care seeking in pregnancy is low considering the low rates of 4 or more antenatal care visits (28%) and facility deliveries (38%).”

Page 6, lines 12-15: maybe revising to “...changing symptoms and behaviors”, as pregnancy is a period characterized also by changes in behaviors, as well.

We have revised accordingly as – “they looked at only non-varying risk factors and did not analyze changing symptoms, behaviors and maternal weight gain throughout pregnancy.”

Page 6, study design: do you have information regarding the % of women who did not consent and their characteristics? Do they differ from those who consented to participate? Also for those women referred by the study staff, are they included in your study or where they excluded? Please clarify.

In the flow chart, we have shown that 123 women refused to consent. This is 0.3% of those eligible. We have added in the Method-Study. Since this was a very low percentage, we did not explore their characteristics.

For the women who were referred by the study staff, they were included in the study since they had consented. We continued to follow them and collect data through the end of the study. The referral was to make sure their morbidities were addressed. We have added this to the methods “Women with fever or elevated blood pressure as measured by study staff were similarly referred for care but continued to be included in the study.”

Page 8, lines 35-42: I understand the argument of less ANC contacts if the pregnancy period is shorter, due to preterm delivery. Nevertheless, ANC is a determinant of birth outcomes. My suggestion would be to include it in the main analysis. There are studies with important findings in LMICs context, related to ANC visits (i.e Pervin J et al. BMJ Open, 2020).

We had many discussions in our group about whether to include ANC visits or not and decided not to for the reasons we describe in the methods. However, we have run the regression analysis with ANC visits in the model and also added in place of delivery. There does not seem to be much difference in the regression coefficients for the other variables, but we now provide the results including ANC visits and place of delivery in the regression. We have provided those regression results in the supplemental tables (S2) and noted this in the results section.

Page 9, lines 27-30: The information of the exclusion of medically indicated preterm birth come very late in the manuscript (Methods section). Please report the % of spontaneous and non-spontaneous preterm deliveries earlier in manuscript (abstract). It would be more accurate to refer to your outcome as “spontaneous preterm birth” throughout the manuscript.

We have now placed the information on spontaneous preterm earlier (see “Outcome Variable” section of Methods). “Preterm births were classified as spontaneous or non-spontaneous (caesarian section or induction), and only spontaneous preterm births were included in the regression analysis.”

In the Outcome sections of Results and abstract, we added “Spontaneous preterm birth was 14.5% and non-spontaneous preterm birth was 0.5%.”

We have changed preterm birth to spontaneous preterm birth, where appropriate, throughout the manuscript, including in the abstract so it is clear at the outset. In the literature, it was not always evident whether the researchers were referring to spontaneous or all preterm.

Page 9, lines 50: how many of the included infants were siblings?

52% women (33% with two pregnancies, 14% with three pregnancies, 4% with four pregnancies and 1% with more than four pregnancies) contributed more than one pregnancy to the study. We have added this to the Results-Participants section.

Page 10, lines 1-10: did you consider multiple imputation or other techniques to treat your missing information, without excluding 29% of your study population? Also did the excluded 29% differ in other characteristics compared to the included women? Since you had such close monitoring and data collection strategy with hired research staff, what was the reason for this missing information?

Most of the missing information was due to late enrollment in the study, in the second or third trimesters, when we first identified women as pregnant. We understand this is a concern (and was also raised by the other reviewer). We have added some description of why data were missing and done an analysis showing differences between those excluded and those included in the regression analysis. We provide this table in the online supplemental materials and describe these differences in the results section. We have added more in discussion on this concern.

Revised in the Statistical Method Section: The descriptive analysis had 31,851 pregnancies. In the regression analysis, we excluded the 1093 pregnancies (3.4%) that ended in caesarian section, induction or both, which leaves 30,758 for analysis. Then, 30.7% out of 30,758 (20.2% missing morbidity in 2nd trimester due to enrollment only in 3rd trimester, 9.4% missing morbidity in 3rd trimester and 1.1% missing other variables) were missing in the regression analysis, and so the final multivariable regression analysis excluded those 9,461 pregnancies, and consisted of 21,297 pregnancies.

Added in the Results-Outcome Section: To examine possible bias associated with exclusion of pregnancies with missing data, we compared characteristics of women excluded in the regression analysis (n=9,461) (mainly because of missing morbidity in 2nd trimester due to late enrollment) with those included in the regression analysis (n=21,297) (supplementary Table S1). The women excluded in the regression analysis were slightly better off than those included in the regression based on education and socioeconomic status but most relevant, the spontaneous preterm prevalence was 17.9% for those excluded in the regression compared to 13.8% included in the regression.

Added in the Discussion Section: Although there was some missing data in regression analyses, a comparison of those with and without missing data did not show large differences in risk factor prevalence. However, those missing data had higher prevalence of preterm birth. It is possible

that if women with missing data were included in the regression, we may have seen stronger associations but the potential bias of these differences is unclear.

You are reporting that “only” 41% was enrolled in the 1st trimester and so you excluded the information collected in the 1st trimester but the same percentage of women were enrolled in the 2nd trimester and you kept the information collected in the 2nd trimester. This is a bit confusing.

You are correct that by 1st trimester only 41% were enrolled. If we kept the information/data of first trimester in the regression as – Did you have GI problem in 1st trimester, only 41% would have replied and 59% would be missing, which would ultimately be missing in the regression analysis. But, by 2nd trimester another 41% enrolled, making it to more than 80% enrollment. So, we kept the 2nd trimester data/variable in the regression analysis.

Table 3: how did you decide on the reference category for each variable?

Our research group discussed and decided on reference categories that made sense based on the variable but we were also guided by using the category with the low risk according to literature of similar settings, to be the reference if there was no clear hierarchy of risk (such as maternal age, caste) but selected the most at risk group for those where a hierarchy existed (such as maternal education, wealth quintile, maternal height). This is added to the Variables section in the paper.

Table 3: what is the unit of the last variable in the table, per kg of body weight?

Yes, the unit of weight is kg. It is now noted in table 3 and also in Table 2

Discussion:

Page 19, lines 15-18: this statement needs context. If excluding the missing information, 66% of the women in your study (82% enrolled at 1st or 2nd trimester) attended 2 or more ANC visits. Do you still consider this as low rate of ANC in an LMIC rural context? What is the comparison for this statement?

In the discussion, we have added “NDHS did not provide data on ANC 4+ for Province 2 but rural areas of Nepal had 62% coverage of ANC 4+. It should also be noted in our study that health care seeking in pregnancy is low considering the low rates of 4 or more antenatal care visits (28%) and facility deliveries (38%).”

Page 19, lines 22-34: I don’t think that the comparisons with US and French populations are adequate. Please use references of studies with similar context as yours.

Meta-analysis done using 14 cohort studies from LMICs and a study from sub-saharan African countries also show that primiparity is associated with increased odds of preterm birth. Primipara is a risk factor for hypertensive disorders of pregnancy (HDP), which increases the risk of preterm birth. This was added to the discussion.

Page 20, lines 15-27: there were several morbidities decreasing the risk for preterm birth, while the discussion of this finding is not adequate. Could the authors consider any reasons for this observation, as for example higher frequently of ANC or care seeking for his group?

Thank You for your suggestion. We have revised this section as- “Pregnancy-varying morbidities that significantly decreased the risk of preterm birth in our analysis were respiratory problems in the 3rd trimester; and poor appetite, nausea and vomiting in the 2nd trimester, and the 3rd trimester. On segregating the symptoms within respiratory problems, we found that it was the persistent cough in the 3rd trimester that decreased the risk of preterm. A similar relationship was found between persistent cough and Large for Gestational Age (LGA) in another study done using the same data as ours. [54] However, we could not find any such association in the previous literature. The association might be due to some unmeasured confounders. Or it could be that women with persistent cough in the 3rd trimester made more frequent check-up visits. We saw that 40% of women with persistent cough in the 3rd trimester sought treatment for cough, and almost all had sought treatment more than once. The pathogenesis of nausea and vomiting in pregnancy is not very clear, but it is broadly accepted to be multifactorial, with the involvement of genetic, endocrine, and gastrointestinal factors. [55] Our findings corroborate with previous findings that nausea and vomiting is associated with reduced risk of preterm birth. [56-59] Specifying by trimesters, a study by Wallin et. al. in Nepal showed similar findings - poor appetite, nausea and vomiting in first trimester was not significantly

associated with spontaneous preterm births, but having these symptoms in the 2nd trimester decreased the risk of spontaneous preterm by 25%. [60]

Page 20, lines 50-51: again the low enrollment rate on 1st trimester is used as an argument for not checking this assumption, even though you have the data and the same enrollment rate as in 2nd trimester. Please clarify.

The clarification is same as in previous point. By 1st trimester only 41% were enrolled. If we kept the information/data of first trimester in the regression as – Did you have GI problem in 1st trimester, only 41% would have replied and 59% would be missing, which would ultimately be missing in the regression analysis. But, by 2nd trimester another 41% enrolled, making it to more than 80% enrollment. So, we kept the 2nd trimester data/variable in the regression analysis.

Page 21, lines 27-29: you are claiming that this is a population of normal BMI but you have not reported that in the tables. Please include this factor in your analysis. In addition, there are more proper metrics to assess weight gain in combination with pre-pregnancy BMI, used internationally.

Actually, this population does not have normal BMI but rather is undernourished. We have added the following to the discussion to clarify this.

Our study was conducted in a non-obese and undernourished population. We do not have pre-pregnancy BMI, so we looked at the mean BMI in the first trimester. Though the first trimester represents less than half of the pregnancies in the study, it hints at undernutrition in the population. The mean BMI was 19.1 kg/m², and 37% had BMI less than 18.5 kg/m². So, less maternal weight gain in such population can pose a risk to spontaneous preterm births.

Limitations and strengths: the use of LMP for GA estimation can produce outcome misclassification bias, if compared to ultrasound-based estimations. Have you considered how big of an issue this is for your study and how it can affect your findings? In addition, the list of factors, important for preterm delivery, that have not been included in this study is much longer than anemia and GDM, like for example maternal nutrition. Please include an adequate assessment of study limitations and how they could have affected your findings.

We have added more to the discussion, and the strengths and limitations sections, to address study limitations, including using LMP rather than ultrasound. Using the same method as we used to obtain LMP, Gernand et al. found that LMP based estimates of GA in rural Bangladesh were a mean 2.8 days longer than what was obtained on ultrasound (A home calendar and recall method of last menstrual period for estimating gestational age in rural Bangladesh: a validation study. Gernand AD, Paul RR, Ullah B, Taher MA, Witter FR, Wu L, Labrique AB, West KP Jr, Christian P. J Health Popul Nutr. 2016 Oct 21;35(1):34. doi: 10.1186/s41043-016-0072-y). We therefore believe that this is probably not a significant limitation.

Some variables associated with increased risk of spontaneous preterm births in previous studies, for example, a prior pregnancy ending in a preterm birth, gestational diabetes, maternal anemia and pre-pregnancy maternal nutritional status were not measured in the main trial. However, other important morbidity variables were measured and used in the analysis. We believe these risk factors are likely generalizable for similar populations in South Asia.

Lastly, even though a rich dataset was collected the authors have not considered that some of the factors collected might interact with each other and different effect estimates can be produced. This is a limitation of the statistical analysis strategy but it can be easily addressed.

You are correct that we have not added interaction variables in the regression models. Even with such a large data set, these can be difficult to detect statistically. We were concerned to add further variables into a regression with so many variables already present. We also did not identify specific interactions that we thought might be important in predicting preterm. If the reviewer has suggestions for us, we would be happy to incorporate these in the model.

VERSION 2 – REVIEW

REVIEWER	Dang, Shaonong Xi'an Jiaotong University Health Science Center
REVIEW RETURNED	06-Oct-2022

GENERAL COMMENTS	Authors have addressed most of my comments, and the manuscript has been improved. However, I still care about adjusting variables, which is key point for association analysis. I suggest that authors should address this issues in the limitaion of discussion part.
------------------	--

REVIEWER	Papadopoulou, Eleni Norwegian Inst Publ Hlth, Department of Environmental Exposures and Epidemiology
REVIEW RETURNED	10-Oct-2022

GENERAL COMMENTS	I have no further comments to the Authors.
------------------	--

VERSION 2 – AUTHOR RESPONSE

Response to Reviewer's comment

Reviewer: 1

Dr. Shaonong Dang, Xi'an Jiaotong University Health Science Center

Comments to the Author:

Authors have addressed most of my comments, and the manuscript has been improved. However, I still care about adjusting variables, which is key point for association analysis. I suggest that authors should address this issues in the limitation of discussion part.

- We appreciate the reviewer is still concerned about why some variables were not included in the final regression model. We have added some further detail about why they were not included in the discussion section under strengths and limitations, as shown here in the italic font.
- Some variables associated with increased risk of spontaneous preterm births in previous studies, for example, a prior pregnancy ending in a preterm birth, gestational diabetes , maternal anemia and pre-pregnancy maternal nutritional status were not measured in the main trial. However, other important morbidity variables were measured and used in the analysis. ***Some covariates were highly correlated with each other (such as some reproductive history ones) and so, not all were included in the multivariable regression. Some covariates were not statistically significant in unadjusted analyses and there was not a compelling biological or sociological reason to include them in the adjusted model. Other important variables like smoking and alcohol, although measured, could not be included in the final regression analysis as their prevalence was very low in this population.*** We believe these risk factors are likely generalizable for similar populations in South Asia.